# Stratification and Adaptation of Malaria Control Interventions in Chad

**DOI:** 10.3390/tropicalmed8090450

**Published:** 2023-09-15

**Authors:** Mahamat Idriss Djaskano, Mady Cissoko, Mahamat Saleh Issakha Diar, Demba Kodindo Israel, Kerah Hinzoumbé Clément, Aicha Mohamed Ali, Makido Dormbaye, Issa Mahamat Souleymane, Adam Batrane, Issaka Sagara

**Affiliations:** 1National Malaria Control Program (NMCP Chad), N’Djamena 1953, Chad; mahamat.idriss@yahoo.fr (M.I.D.); saleh_diar@yahoo.fr (M.S.I.D.); iskodindo@yahoo.fr (D.K.I.); makidod@gmail.com (M.D.); issa.mahamatsoul@gmail.com (I.M.S.); batrami10@yahoo.fr (A.B.); 2Malaria Research and Training Center, FMOS-FAPH, University of Sciences, Techniques and Technologies of Bamako, Bamako BP 1805, Mali; madycissoko@ymail.com; 3SESSTIM, UM1252, ISSPAM, INSERM, IRD, Aix Marseille University, 13005 Marseille, France; 4United Nations Development Program (UNDP), Support Project for Malaria Control in Chad (PA-LAT), N’Djamena BP 906, Chad; clement.hinzoumbe@undp.org (K.H.C.); aicha.mohamed.ali@undp.org (A.M.A.)

**Keywords:** stratification, malaria, interventions, environnent, Chad

## Abstract

Malaria remains the leading cause of morbidity and mortality in Chad. The World Health Organization (WHO) has recommended that endemic countries stratify malaria to guide interventions. Thus, the Republic of Chad has initiated a stratification process based on malaria incidence with the aim of defining transmission risk and proposing interventions. We collected routine malaria data from health facilities from 2017–2021, the national survey on malaria indicators, the entomological data of NMCP operational research, the demographic and health surveys, and remote sensing of environmental data. Stratification was based on the adjusted incidence of malaria to guide interventions. The adjusted incidence of malaria was, on average, 374 cases per 1000 people in the country. However, it varied according to health districts. Health districts were stratified into very low malaria incidence (*n* = 25), low malaria incidence (*n* = 20), moderate malaria incidence *(n* = 46) and high malaria incidence *(n* = 38). Micro-stratification in health districts with very low incidence was carried out to identify districts with incidence <10 cases per 1000 person with a view to a malaria pre-elimination programme. Appropriate malaria control interventions were proposed based on the strata identified. Stratification enables the country to target interventions to accelerate the reduction of the burden caused by malaria with a pre-elimination goal.

## 1. Introduction

Malaria is a major public health problem worldwide, particularly in sub-Saharan Africa. It is caused by a hematozoan of the genus *Plasmodium*, transmitted to humans by the bite of an infected female mosquito of the genus *Anopheles* during its blood meal [1]. The WHO has reported approximately 241 million cases and 627,000 deaths from malaria in 2020 in the world [2], with most cases (93%) and deaths (96%) in the African region. Children under five (5) years old are the most vulnerable to malaria with 80% of deaths in 2020 in Africa.

Malaria remains the main cause of morbidity and mortality in Sub-Saharan African countries, particularly Chad. In Chad, malaria is endemic in areas at risk of epidemics. It represents the first reason for consultation (42%), hospitalization (32%) and hospital deaths (30%). Children under five (5) years old are the most affected (43%)(Unpublished document). Malaria is as prevalent as 40.9% in the general population and 41% in children under five (5) years old [3]. The National Malaria Control Program (NMCP) database reported 2,747,020 suspected malaria cases out of 4,876,425 cases of all combined pathologies in health facilities in 2021. Among the suspected cases, 2,228,512 (81%) were tested biologically, with 1,548,039 (69.46%) confirmed positive. Malaria accounted for 3303 (30%) deaths out of all hospital deaths (n = 11,099).

The 2017 National Survey on Malaria Indicators in Chad (SMI) has reported three (3) types of *plasmodium* in the country: *Plasmodium falciparum* (85.1%), *Plasmodium malariae* (20.9%) and *Plasmodium ovale* (6.4%) with two new subspecies *Plasmodium wallikeri* and *Plasmodium curtisi* [3].

With regard to vector species, out of about fifteen species of *Anopheles* listed in the sentinel sites, only five are involved in the transmission of malaria (*Anopheles arabiensis*, *An. gambiae ss*, *An. funestus*, *An. pharoensis* and *An. ziemanni*). *An. arabiensis* and *An. gambiae ss* are the most widely distributed and account for nearly 85% of transmission. Following evaluations of the susceptibility to insecticides used in public health, highly variable responses have been obtained for *An. gambiae sl* [4,5,6].

In Chad, the fight against malaria is coordinated by the National Malaria Control Program (NMCP), with a mission to develop policies, standards, strategies, and guidelines for the fight against malaria. This policy is broken down into the National Strategic Plans (PSN). The latest PSN covers 2019–2023 and provides interventions with proven efficacy for malaria prevention and management. The implementation of all the interventions was based on the epidemiological profile established over forty (40) years ago.

In Chad, considering the dynamics of transmission and the level of endemicity, malaria was categorized according to three epidemiological facies corresponding to three distinct geo-climatic zones: (i) the desert north, free from malaria transmission according to the available data, (ii) the central Sahelo-Saharan, corresponds to unstable malaria, due to a shorter malaria transmission season (<three months) and, (iii) the southern Sudan characterized by stable malaria with a quasi-seasonal and long (4 to 6 months) malaria transmission season.

Some interventions, such as Seasonal Malaria Chemoprevention (SMC), were not carried out throughout high malaria transmission for each health district. All the districts of the Sahelian zone only benefited from the SMC but with four (4) cycles without exception. Districts in the Sudanian zone were not included in the SMC programme. However, this stratification, which dates back more than 40 years, requires updating, and interventions must be adapted to the current epidemiological profile of malaria. Stratification is the classification of geographical areas or localities according to epidemiological, ecological, social, and economic determinants to guide antimalarial interventions [7].

This study aims to define the risk of transmission and propose adaptations of malaria control measures, such as Seasonal Malaria Chemoprevention based on the incidence of malaria.

## 2. Materials and Methods

### 2.1. Study Site

The study occurred in Chad, a country in the heart of the African continent, between the 7th–24th degree of North latitude and the 13th–24th degree of East longitude. Covering an area of 1,284,000 km^2^, it is bounded to the North by Libya, the South by the Republic of Central Africa, the West by Cameroon, Nigeria, and Niger, and the East by Sudan. Chad had 18 million inhabitants in 2021. The northern third of the country is part of the Sahara Desert. It rains there rarely and irregularly, with rainfall of less than 200 mm (mm) per year. The climate is harsh and not conducive to agriculture. The density is very low, with about one inhabitant per km^2^ [8]. In the center is the Sahelian steppe, where the rainfall varies between 200 and 800 mm per year. The South is a flat and very clayey region. When it rains, this tropical region becomes a huge swamp, making it impossible to move around. The vegetation is abundant. The average rainfall is 800 to 1200 mm per year [9]. Chad has 129 health districts across the three geo-climatic zones.

### 2.2. Data Collection

#### 2.2.1. Malaria Data

We extracted data on confirmed malaria cases from 2017 to 2021 from the NMCP database, where data reported by health facilities are stored. We collected data on the prevalence of malaria and parasitic species through the 2017 SMI and infant and child mortality rates through the Demographic and Health Province Surveys in Chad (EDST) from 2004 to 2019.

#### 2.2.2. Environmental Data

Without exhaustive rainfall data, rainfall data were collected through remote sensing from NASA’s GIOVANNI site (satellite data: giovanni.gsfc.nasa.gov/) accessed on 22 October 2022. Daily rainfall data in mm were collected at the health district level from 2017 to 2021.

#### 2.2.3. Population Data

The population data was obtained through the General Census of Population and Housing 2009, updated annually by the Department of Health Information System (DSIS) in collaboration with the National Institute of Statistics, Economic and Demographic Studies (INSEED).

#### 2.2.4. Data Quality Checking

These data are validated every six months by all those involved in the fight against malaria. Data consistency was achieved through field visits to 66 health facilities. This visit consisted of counting all the registered cases and comparing them with the data reported by the month of the second half of 2021 (period of high malaria transmission). This helped to determine the rate of data discrepancy between the health facility registries and the Monthly Malaria Report (MMR).

We determined the rates for each variable, and we then averaged out by province using this formula:-Register and MMR data mismatch rate = ×100-Data agreement rate = 100 − discrepancy rate

We assessed the concordance/consistency/accuracy rate of the data between the registries and the MMR as follows:-greater than 95% was labeled good data quality;-between 90 to 95% was labeled average data quality;-less than 90% was labeled poor data quality.

#### 2.2.5. Data Analysis

##### Annual Incidence of Malaria

This was the number of confirmed malaria cases in a year reported to the 1000 inhabitants in the general population. The incidence of malaria was determined and adjusted by considering the rates of confirmation of biological test diagnosis and attendance at health facilities for each health district. This made it possible to determine four (4) scenarios of incidence:

##### Scenario 1: Crude Incidence

The crude incidence was determined by reporting the cases of malaria confirmed by biological tests per 1000 inhabitants in the general population.

##### Scenario 2: Incidence Adjusted for Biological Confirmation Rate

Since not all suspected malaria cases were tested for biological confirmation, we assumed that this crude incidence calculated based on confirmed malaria cases did not reflect the true situation. To this end, we made an adjustment using the biological confirmation rate.

The rate of biological confirmation per health district was calculated by dividing confirmed cases by the number of cases tested by 100.

The WHO recommends to countries that all suspected cases of malaria should be diagnosed parasitologically before treatment is started. All suspected cases must be tested at 100% (suspected cases = tested cases). We, therefore, applied the biological confirmation rate to suspected cases to estimate the adjusted cases. These adjusted cases per 1000 of the population were used to determine the incidence adjusted for the biological confirmation rate.
adjusted cases=suspected cases×biological confirmation rate
$$adjusted\;incidence=\frac{adjusted\;cases}{population}\times 1000$$

##### Scenario 3: Incidence Adjusted for Health Facility Attendance Rate

The attendance rate was determined for each health district by dividing the total number of consultations for all diseases over the population by 100. This rate was used to calculate the adjusted cases per health district.

Cases adjusted to the attendance rate were determined by dividing confirmed cases by the attendance rate. These adjusted cases, divided by the population per 1000 person-years, were used to determine the incidence adjusted for the attendance rate.
$$Attendance\;rate=\frac{Total\;consultations}{population}\times 100$$
$$Adjusted\;cases=\frac{confirmed\;case}{Attendance\;rate}$$
$$Adjusted\;incidence=\frac{Adjusted\;cases\;}{population}\times 1000$$

##### Scenario 4: Incidence Adjusted for Laboratory Diagnosis Confirmation and Attendance Rates

The incidence adjusted for the biological confirmation rate and the attendance rate was calculated as follows:

Firstly, cases adjusted to the biological confirmation rate and the attendance rate were calculated by dividing cases adjusted to the biological confirmation rate by the attendance rate. These adjusted cases per 1000 population were used to determine the incidence adjusted for the biological confirmation rate and the attendance rate.
$$cases\;adjusted\;to\;confir\;and\;freq\;rates=\frac{cases\;adjusted\;to\;biological\;confir\;rate}{Attendance\;rate}$$
$$adjusted\;incidence=\frac{adjusted\;cases\;confir\;and\;freq\;rate}{population}\times 1000$$

Risk classes (strata) were defined based on incidence adjusted for laboratory confirmation and attendance rates based on the classification of transmission areas of the WHO Malaria Elimination Framework [7]:-Very low malaria transmission zone: incidence less than 100 cases per 1000 person (or parasite prevalence < 1%).-Low malaria transmission zone: incidence between 100 to 250 cases per 1000 person (or parasite prevalence 1–10%);-Moderate malaria transmission zone: incidence between 250 to 450 cases per 1000 person (or parasite prevalence between 10–35%);-High malaria transmission zone: incidence greater than 450 cases per 1000 person (or parasite prevalence > 35%).

The very low incidence zone was subdivided into three (3) classes (<10 cases per 1000, between 10 to <50 cases per 1000, and between 50 to <100 cases per 1000) to target better malaria interventions.

##### Prevalence of Malaria

We calculated the prevalence of malaria infection in the general population and in children aged 6–59 months old based on the results of the 2017 SMI at the provincial level. It was obtained by calculating the number of people with a positive malaria result to the number of people in the general population tested during the survey.

##### Distribution of Malaria Parasites

The distribution of parasite species was determined by province based on the results of the 2017 SMI.

##### Infant and Child Mortality

We determined Infant and child mortality based on the 2004 and 2019 EDSTs results. We calculated the infant and child mortality rate by comparing children who died before the age of five (5) years old to the entire population of this same age group for a year. It was expressed per 1000 (%).

##### Meteorological Data

We determined the average monthly rainfall for the last five (5) years (2017–2021) at the health district level.

##### Entomology

We determined the distribution of vector species and the monitoring of vector resistance to insecticides in the sentinel sites based on the results of studies conducted by the NMCP from 2010 to 2021. We conducted sensitivity tests to insecticides on the *Anopheles gambiae complex*, a major vector of malaria in Chad, using eight (8) insecticides belonging to the four (4) chemical families used in public health. The results were interpreted as follows:-Mortality < 90%: Resistance confirmed;-Mortality between 90–97%: Probable resistance;-Mortality between 98–100%: Susceptible.

##### Seasonality of Malaria

The seasonality of malaria was studied through the monthly incidence time series from 2017 to 2021. The time series was split into periods of high malaria transmission versus low malaria transmission using *ChangePoint* Analysis for five (5) years. This made it possible to determine the beginning and end of the period of high malaria transmission by the health district. We calculated medians for the start and end of the high transmission period for each district. We determined the duration of the high transmission period by taking the difference between the two medians (end and start) for each health district to adapt control interventions such as SMC.

##### Targeting of Interventions by Transmission Zone

The targeting of interventions is part of accelerating the reduction of the burden of malaria provided for in the National Strategic Plan for the fight against malaria in Chad. It aligns with the WHO’s Global Technical Strategy for Malaria Control 2016–2030 [9]. To do this, control actions will focus on targeting interventions according to the risk of transmission, seasonality, accessibility to health services, infant and child mortality, vector resistance to insecticides and environmental factors.

#### 2.2.6. Statistical Tools

We performed statistical analysis of the data on R software version 4.2.1 (The R Foundation for Statistical Computing, 2022). We used QGIS software version 3.4 (QGIS Development Team, 2022, QGIS Geographic Information System, Open-Source Geospatial Foundation Project. http://qgis.osgeo.org (accessed on 19 October 2022)) to produce the incidence, prevalence, mortality maps, parasite species distribution, malaria vector mapping, vector resistance to insecticides and rainfall.

#### 2.2.7. Ethical Considerations

We did not involve data on individuals directly. Instead, we analyzed historical data, mainly from the aggregated anonymous database, after we obtained authorization from the NMCP for their use (Authorization N°642/PR/PM/MSP/DG/NMCP/2019 of 7 November 2019).

## 3. Results

### 3.1. Malaria Incidence

A total of 1,548,039 malaria cases were recorded in Chad’s 129 health districts in 2021. The country’s annual crude incidence of malaria was 85.42 per 1000, with variation between districts. We observed the lowest incidence in the Kouba-Olanga district, with 0.24 cases per 1000 and the highest in the Moissala district, with 684 cases per 1000.

This adjusted incidence was 157 cases per 1000 in the whole country, varying from 6 to 1494 cases per 1000 between the districts of Zouar and Koumogo.

### 3.2. Malaria Crude Incidence

Crude annual malaria incidence per 1000 persons at the district level in 2021 was provided.

Out of 129 health districts covered by the study, the crude annual incidence of malaria (Figure 1) according to the WHO classification was as follows:-One hundred and two (102) health districts are in the very low malaria transmission zone;-Nineteen (19) health districts are in the low malaria transmission zone;-Six (6) health districts are in the moderate malaria transmission zone;-Two (2) health districts are in the high malaria transmission zone.

### 3.3. Incidence Adjusted for Malaria Laboratory Confirmation Rate

The annual incidence of malaria adjusted to the biological confirmation rate (Figure 2) according to the WHO classification was as follows:-Eighty-seven (87) health districts are in the very low malaria transmission zone;-Twenty-nine (29) health districts are in the low malaria transmission zone;-Eleven (11) health districts are in the moderate malaria transmission zone;-Two (2) health districts are in the high malaria transmission zone.

### 3.4. Incidence Adjusted for Attendance Rate

The annual incidence of malaria adjusted to the attendance rate according to the WHO classification was as follows (Figure 3):-Twenty-eight (28) health districts are in the very low malaria transmission zone;-Thirty-nine (39) health districts are in the low malaria transmission zone;-Forty-one (41) health districts are in the moderate malaria transmission zone;-Twenty-one (21) health districts are in the high malaria transmission zone.

#### Accessibility to Health Services

Figure 3: Attendance map of health districts in 2021.

The average health service attendance rate was 28%, which varied according to the health districts (Figure 3). The lowest rate was observed in the health district of Kouba Olanga, 4.7% and the highest in Goré, with 74.6%.

### 3.5. Incidence Adjusted for Malaria Laboratory Confirmation and Attendance Rates

Figure 1: Annual malaria incidence adjusted for biological confirmation and attendance rates per 1000 persons by district in Chad in 2021.

The annual incidence of malaria adjusted for laboratory confirmation and attendance rates according to the WHO classification is as follows (Figure 4):-Twenty-five (25) health districts are in the very low malaria transmission zone, including five (5) districts with an incidence <10 per 1000, nine (9) districts with an incidence between 10 and <50 per 1000 and 11 districts with an incidence between 50 and <100 cases per 1000 person;-Twenty (20) health districts are in the low malaria transmission zone;-Forty-six (46) health districts are in the moderate malaria transmission zone;-Thirty-eight (38) health districts are in the high malaria transmission zone.

### 3.6. Incidence of Malaria in Children under Five (5) Years Old

The incidence adjusted for laboratory confirmation and malaria attendance rates per 1000 person-years in children under five (5) years old was 788 cases per 1000 children nationwide but varied by district. The lowest incidence was observed in Zouar, with six (6) cases per 1000 people, and the highest was in Moissala, with 3909 cases per 1000 people.

### 3.7. The Prevalence of Malaria

The prevalence of malaria infection in the general population and in children aged 6–59 months old in 2017 according to the SMI results.

Figure 4 and Figure 5: Malaria prevalence in the general population and children aged 6–59 months by province in Chad in 2017.

The average prevalence of malaria infection in the general population (Figure 4) is 40.8% with a disparity according to the provinces. It is low (1.6%) in the provinces of the Sahelo-Saharan zone of the country (Lac, Kanem and Mao), intermediate (15.9%) in the provinces of the Sahelian zone of the country (Guera, Batha and Hadjer Lamis) and high (82.2%) in the Sudanian zone of the country (Logone Occidental (Logone Oriental and Tandjilé). The average prevalence of malaria infection in children aged 6–59 months (Figure 5) old was 40.9%, with variation by province. According to the transmission zones, the results showed a higher prevalence (80.3%) in the provinces of the Sudanian zone (Logone Occidental, Logone Oriental and Tandjilé). In the provinces of the Sahelian zone of the country (Guera, Batha and Hadjer Lamis), the prevalence was 18.8%. It was very low (1.3%) in the provinces of the Sahelo-Saharan zone of the country (Lac, Kanem and Mao).

### 3.8. Distribution of Parasitic Species in Chad

The distribution of malaria parasite species by province from 2017 SMI was summarized (Figure 6).

According to the results of the 2017 SMI on molecular analyses, *Plasmodium falciparum* remains the most common plasmodial species in the country (85.1%), followed by *P. malariae* (20.9%) and *P. ovale* (6.4%). This study described the presence of two new subspecies of Plasmodium ovale (wallikeri and curtisi). The results showed that all three plasmodial species are present throughout the country except Barh El-Gazel, Batha, Hadjer Lamis, Kanem, Lac, Ouaddai and Wadifira. In the provinces of Barh El-Gazel, Ouaddai and Wadi-Fira, *P. falciparum* has been found without association with other species (Figure 6).

### 3.9. Infant and Child Mortality

The infant and child mortality data collected during the EDST survey in 2004 and 2019 were presented (Figure 7 and Figure 8).

Figure 7 and Figure 8: Infant and child mortality in 2004 and 2019 in Chad.

In 2004, the infant and child mortality rate was over 100 deaths per 1000 live births in all the country’s provinces (Figure 7). On the one hand, in 2019, we noted that only ten provinces out of 23 still have this extreme value (Figure 8). The lowest infant and child mortality rates were observed in the provinces of Borkou, Tibesti, Wadi-Fira and Ennedi Est, respectively, 33, 33, 38 and 50 per 1000 live births. On the other hand, Chari Baguirmi, Logone Oriental and Logone Occidental recorded the highest rates, 179, 188 and 197 per 1000 live births. The provinces with high infant and child mortality (>100 per 1000) were located in the Sudanian zone of the country, corresponding to the zone of high malaria transmission. Infant and child mortality fell from 203 per 1000 live births in 2004 to 122 per 1000 live births in 2019, a reduction of 60.09% (Figure 8).

### 3.10. Weather Factors from 2017 to 2021

#### 3.10.1. Rainfall Situation by Health District

The average annual rainfall by health district is summarized (Figure 9).

Figure 9: Annual average rainfall map over five (5) years (2017–2021) of health districts.

A 5-year average rainfall (2017–2021) measured ranged from less than 200 mm in the Saharan zone to more than 800 mm in the Sudanian zone. Of the 129 health districts in Chad, fourteen (14) districts had a rainfall < 200 mm, corresponding to the Saharan zone, thirty-seven (37) districts had a rainfall between 200 to 600 mm (Sahelo-Saharan zone), sixteen (16) districts had a rainfall between 600 to 800 mm (Sahelian zone), 51 districts have rainfall between 800 to 1200 mm corresponding to the Sudanian zone and 11 districts have a rainfall > 1200 mm corresponding to the Sudano-Guinean zone (Figure 9).

#### 3.10.2. Health Districts and Period of High Transmission

The period of high malaria transmission in the health districts varied between 0 to 7 months. It matched with the different geo-climatic zones in the country. In the Sudano-Guinean zone, it was six (6) to (7) months. In the Sudanian zone, it was five (5) months. In the Sahelian zone, it was four (4) months; in the Sahelo-Saharan zone, it hardly exceeded three (3) months. In the Saharan zone, there is no specific period of high transmission. It rarely rains (Figure 10).

##### Estimation of Malaria Seasonality in Districts

The analysis of the time series of the incidence of malaria over the five years (2017–2021) using Change Point Analysis made it possible to map the health districts with the period of high and low malaria transmission as well as the length of the transmission season:-Five (5) health districts have two (2) months of high malaria transmission period;-Nineteen (19) health districts with three (3) months of high malaria transmission period;-Thirty-two (32) health districts with four (4) months of high malaria transmission period;-Forty-four (44) health districts have five (5) months of high malaria transmission period;-Nine (9) health districts have six (6) months of high malaria transmission period;-Two (2) health districts with seven (7) months of high malaria transmission period.

### 3.11. Targeting of Interventions by Stratum

Suggested targeted interventions were provided according to the risk of malaria, seasonality, parasite prevalence, infant and child mortality, vector resistance to insecticides and environmental risk (Table 1). Given the persistence of malaria transmission throughout the country, some interventions should be continued. Biological diagnosis and treatment of clinical cases should be continued. Community-based malaria care should be strengthened in districts with high and moderate incidences of malaria and districts with low attendance rates. Intermittent preventive treatment in pregnant women must be carried out in all malaria transmission areas.

Long-Lasting Insecticide Impregnated Mosquito Net (standard) (LLIN) remains the main vector control intervention. It should be continued and distributed in the countryside, and routinely according to the following modalities: (i) districts with an incidence of less than 50 cases per 1000 should not require LLINs in the countryside, but the distribution should be continued routinely to children under one year and pregnant women. LLIN type should be standard. (ii) For districts where proof of susceptibility to insecticides is approved, they should continue with standard IRS and LLINs routinely and in the campaign. Areas that have not benefited from a test to monitor vector resistance to insecticides should continue the distribution of standard LLINs. Monitoring of these insecticides is strongly recommended. Areas where vector resistance to insecticides has been approved should use LLINs in the campaign and routinely use vector-sensitive insecticides.

SMC in children under five (5) years old should be done in areas (high, moderate, and low incidence) where malaria transmission is seasonal. Districts with very low malaria incidence (<50 cases per 1000) and/or showing no seasonality should be excluded from SMC. The number of passages should depend on the duration of high malaria transmission. Districts with a transmission period of 5 months should make five rounds, those with a transmission period of 4 months should make four rounds, and those with a transmission period of 3 months should make three rounds. Districts with a transmission duration greater than five months and presenting seasonality should have at least five rounds. They must be the subject of an additional study to determine the ideal month to start SMC.

Indoor residential spraying should be carried out in districts with high malaria transmission to reduce disease-related morbidity and mortality. Districts with very low and low malaria transmission should do IRS as an outbreak response.

### 3.12. Health Districts Eligible for SMC and Number of Visits

The number of SMC cycles depends on the duration of high malaria transmission. The map above shows the health districts eligible for SMC and the number of months of cycles (Figure 11). The health districts colored in white did not show the seasonality of malaria. Districts in blue were eligible for SMC with three (3) months of cycles, in yellow with four (4) months of cycles, and in red with at least five (5) months of cycles.

### 3.13. Districts Eligible for SMC and Start-Up Months

The start of SMC depends on the start of the period of high malaria transmission. The health districts colored in blue started SMC in June, those in yellow started in July, and those in red started in August. For health districts with a very long malaria transmission seasonality (more than five months), it was recommended to carry out a study in these health districts with more than five (5) months of high malaria transmission period to find the timely month for starting SMC (Figure 12).

## 4. Discussion

### 4.1. Malaria Incidence

Stratification analysis based on crude malaria incidence in Chad in 2021 showed that almost all districts except two (102/129) were in the very low malaria transmission class (Figure 1). Those two districts were in the high malaria transmission class. Cissoko M. et al., in 2019 in Mali, analyzing the crude incidence of malaria according to the WHO classification, have found a similar result. No district in Mali was in the area of high malaria transmission. These authors used incidence adjusted for attendance in their study to determine the risk of malaria. In our case, the biological diagnostic confirmation test rate and the health facility attendance rate were used. They showed that almost a third of the districts (38/129) were in the area of high malaria transmission (Figure 2). To overcome this problem of shortage of antimalarial inputs in health facilities so that all suspected cases of malaria can benefit from biological diagnostic tests and in accordance with the national policy for the fight against malaria in Chad and the recommendations of WHO [9], a quantification of inputs based on the incidence adjusted to the rates of laboratory diagnostic confirmation test and attendance of malaria would be strongly recommended to the malaria control program in Chad. To facilitate access to care by the population, the extension of community care in areas with high and moderate transmission with a low attendance rate (<40%) is strongly recommended [10].

The transmission of malaria in Chad varies according to the health districts and is linked to bioclimatic conditions (dry and rainy seasons) and hydrogeography [11]. This variation in malaria transmission by the district over two periods and linked to environmental factors has also been noted in Mali [12,13,14,15]. In some countries like Burkina-Faso, malaria transmission was characterized by three transmission periods, one of which was intermediate [11].

### 4.2. Incidence of Malaria in Children under Five (5) Years Old

The incidence adjusted for laboratory diagnostic test confirmation and malaria attendance rates per 1000 person-years in children under five (5) years old was 788 cases per 1000 children nationwide, varying by district. The lowest incidence was observed in Zouar, with six (6) cases per 1000 people, and the highest was in Moissala, with 3909 cases per 1000 people. These results show that children under five (5) years old in the district of Moissala (south of the country) each had at least three (3) episodes of malaria in 2021. This confirms that children under five (5) years old are known to be more vulnerable to malaria [16].

### 4.3. Prevalence of Malaria in Children Aged 6–59 Months Old

It was, on average, 40.9% with a variation by province (Figure 4): Lac 1.3%, Ennedi Est 6.8%, Batha 18.8%, Chari-Baguirmi 37.4%, Mayo-Kebbi Est 50.9%, Mandoul 68.4 and Tandjilé 80.3%. This disparity in the prevalence of malaria was linked to the different geo-climatic zones of the country corresponding to the different zones of malaria transmission. These results were comparable to those of the 2018 EDST VI of Mali [17], which gave an average prevalence of malaria of 18.9% in children aged 6–59 months old. This prevalence also varied according to the regions: Kidal, Taoudenit, Timbuktu between 1% and 10%, Gao and Ménaka above 10%, Sikasso 29.7%, Ségou 25.9% and Mopti 24.9%. The Survey of Malaria Indicators in Senegal in 2020–2021 [18] revealed only an average prevalence among children aged 6–59 months old of 12.5%, with a variation of 18.1% in the regions of South (Kedougou) to 5% in the western regions (Tambacounda, Kolda and Kédougou).

### 4.4. Infant and Child Mortality

According to the 2004 and 2019 EDSTs, infant and child mortality in Chad fell from 203 cases per 1000 live births in 2004 to 122 cases per 1000 live births in 2019, i.e., a reduction of 60.1% in 15 years (Figure 7). This decline in infant and child mortality over the past 15 years could be explained by the various preventive and curative strategies implemented in children under five (5) years old on the one hand and by improving maternal health on the other hand. These include immunization, breastfeeding, nutritional program, use of LLINs, SMC, postnatal care, treatment of common childhood illnesses (malaria, diarrhoea, pneumonia), improvement of maternal health through prenatal consultation, intermittent preventive treatment during pregnancy and assisted delivery. This decline was also observed in Mali during the DHS of 1987 and 2018 [18], i.e., 247 and 101 cases per 1000 live births, i.e., a reduction of 41% in 31 years. Burkina-Faso also recorded a reduction in the infant and child mortality rate between 1993 (187‰ live births) and 2019 (87.3‰ live births), i.e., a reduction of 47% in 26 years [19].

### 4.5. Interventions by Stratum

In the area where malaria incidence was less than 10 cases per 1000 person, with very few risk factors, the control strategy should be based on an epidemiological surveillance system to detect epidemics and molecular surveillance [20,21]. These very low transmission districts could be potential candidates for malaria pre-elimination. In the zone with high and moderate transmission, the control strategy must be based on universal access to prevention and treatment (Table 1). Prevention through CDM and IRS must be adapted to the level of vector resistance to insecticides [22,23].

Community-based malaria case management in Chad began in 2018 in two provinces with the highest malaria incidence. In malaria-endemic countries, most episodes were treated mainly at home and not in public health establishments which were not frequented by most of the population because of problems of geographical accessibility, low economic capacity, and socio-cultural considerations. This was why improving home treatment of uncomplicated malaria cases has been identified as a key strategy to achieve the Roll Back Malaria (RBM) targets in countries [7]. In Africa, over 70% of malaria episodes in rural areas and over 50% in urban areas are self-medicated [24,25]. The informal sector offers inappropriate drugs and unsuitable treatments for the management of malaria. Given the importance of self-medication and the fear that drug resistance will develop due to the indiscriminate use of antimalarials, strategies to improve self-medication are among the cornerstones of successfully implementing the CPR community [24,25].

### 4.6. Seasonal Malaria Chemoprophylaxis (SMC)

It was implemented in 2013 in the Sahelian zone with a maximum of four (4) visits. The start of the SMC was done in the 7th month (July) of the year in all the districts. The analysis of the results of this study showed that the start of the SMC could be done according to the beginning of the period of high malaria transmission. Some districts could start in June, others in July and August (Figure 12). In Mali, a study found that administering an additional round of SMC in addition to four (4) rounds in an area of long seasonal transmission would protect children against malaria by more than 50% [26]. The feasibility and effectiveness of SMC must use contextualized approaches to adapt to local conditions during implementation, which requires a significant level of coverage. It is, therefore, necessary to examine the different possible approaches in the context of Nigeria to put in place effective distribution systems for an extension of the intervention to areas of northern Nigeria with high transmission of seasonal malaria [27]. The WHO guide also recommends that each country adapt the number of SMC cycles to the seasonality of malaria [28].

## 5. Conclusions

Stratification of malaria was done based on incidence adjusted for biological confirmation rate and health facility attendance rate. The study showed that 29% of districts are in the very high incidence zone, 36% in the moderate incidence zone, 15% in the low incidence zone and 19% in the very low incidence zone. Targeted interventions were proposed according to the different transmission zones and considering malaria risk, seasonality, vector resistance to insecticides and environmental risk.

## 6. Ethical Considerations

We did not involve data on individuals directly. Instead, we analyzed historical data, mainly from the aggregated anonymous database, after we obtained authorization from the NMCP for their use (Authorization N°642/PR/PM/MSP/DG/NMCP/2019 of 7 November 2019).

## Figures and Tables

**Figure 1 tropicalmed-08-00450-f001:**
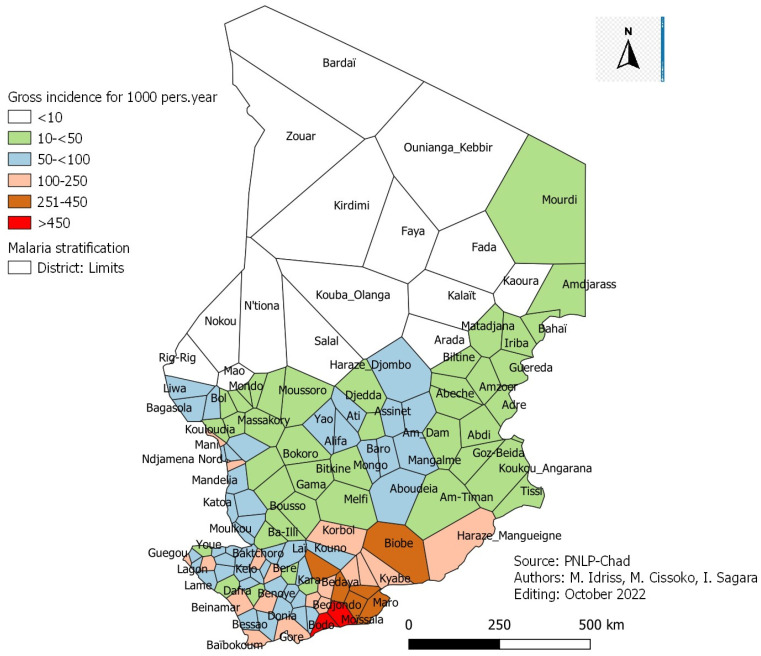
Crude annual malaria incidence per 1000 persons at the district level.

**Figure 2 tropicalmed-08-00450-f002:**
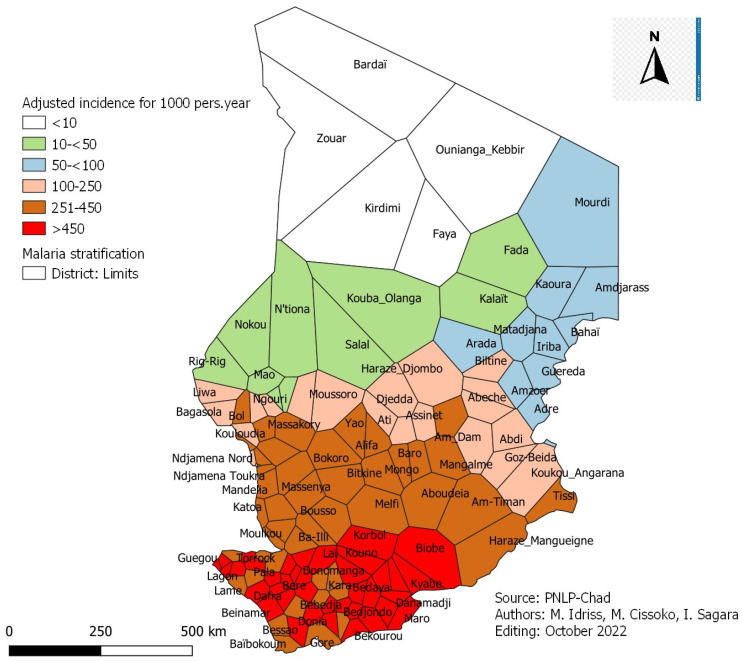
Annual malaria incidence adjusted for biological confirmation and attendance rates per 1000 person by district in Chad in 2021.

**Figure 3 tropicalmed-08-00450-f003:**
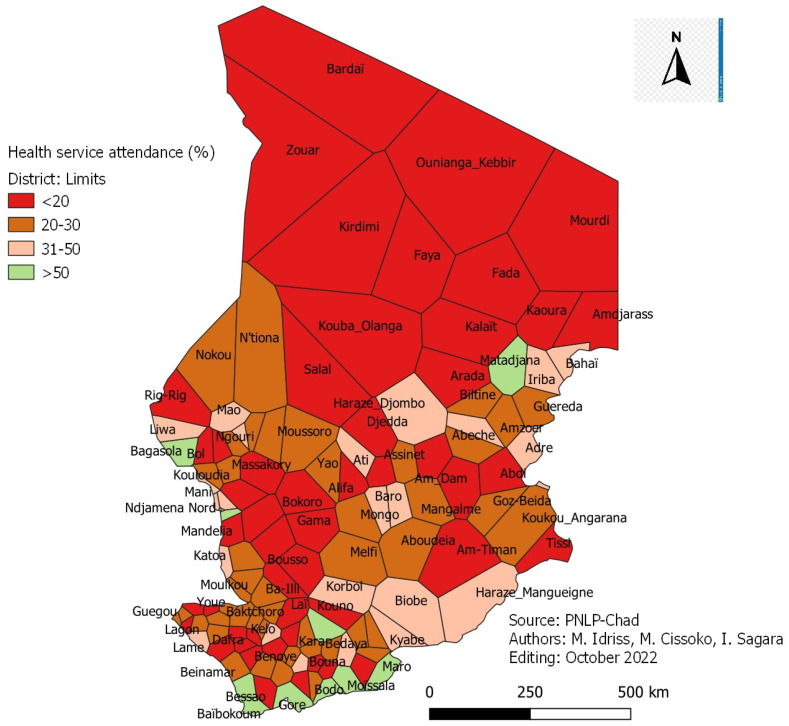
Attendance map of health districts in 2021.

**Figure 4 tropicalmed-08-00450-f004:**
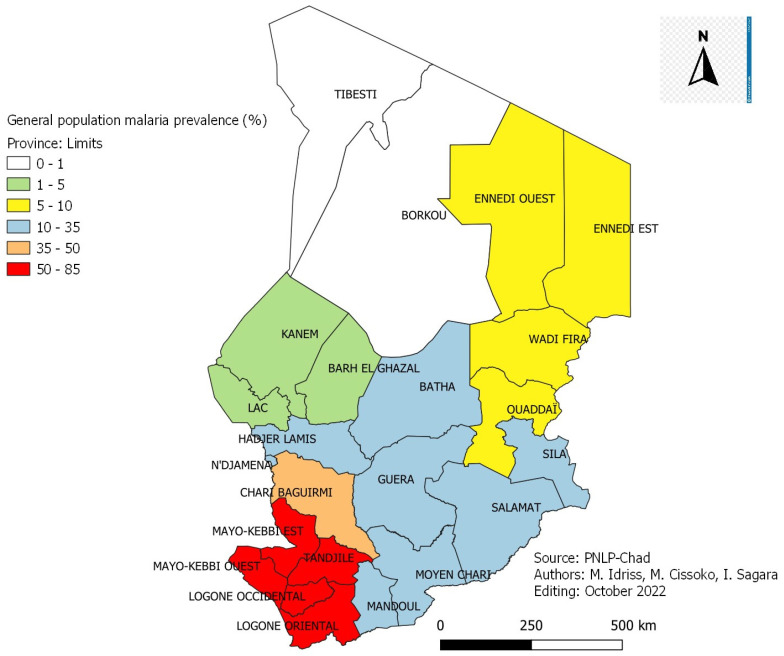
Malaria prevalence in the general population by province in Chad in 2017.

**Figure 5 tropicalmed-08-00450-f005:**
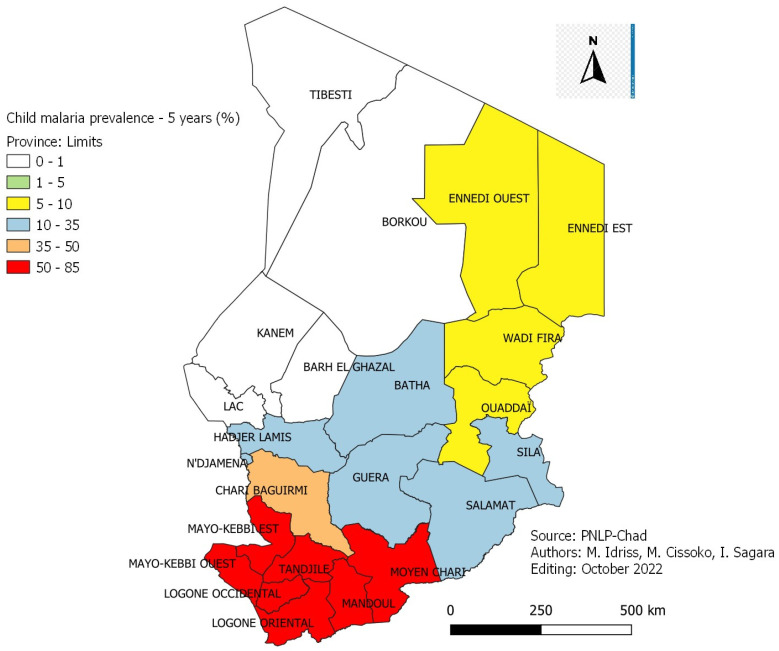
Malaria prevalence in children aged 6–59 months by province in Chad in 2017.

**Figure 6 tropicalmed-08-00450-f006:**
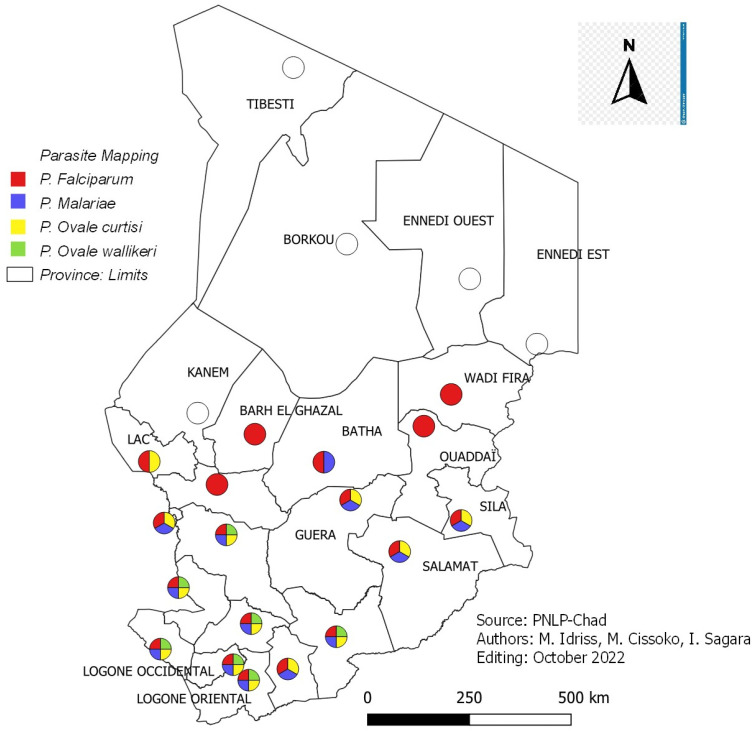
Parasite mapping by the province in 2017.

**Figure 7 tropicalmed-08-00450-f007:**
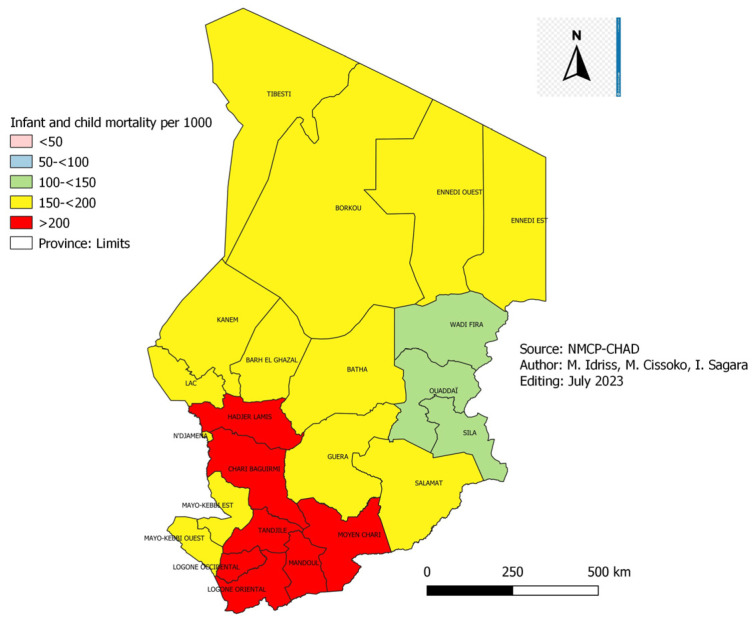
Infant and child mortality in 2004 in Chad.

**Figure 8 tropicalmed-08-00450-f008:**
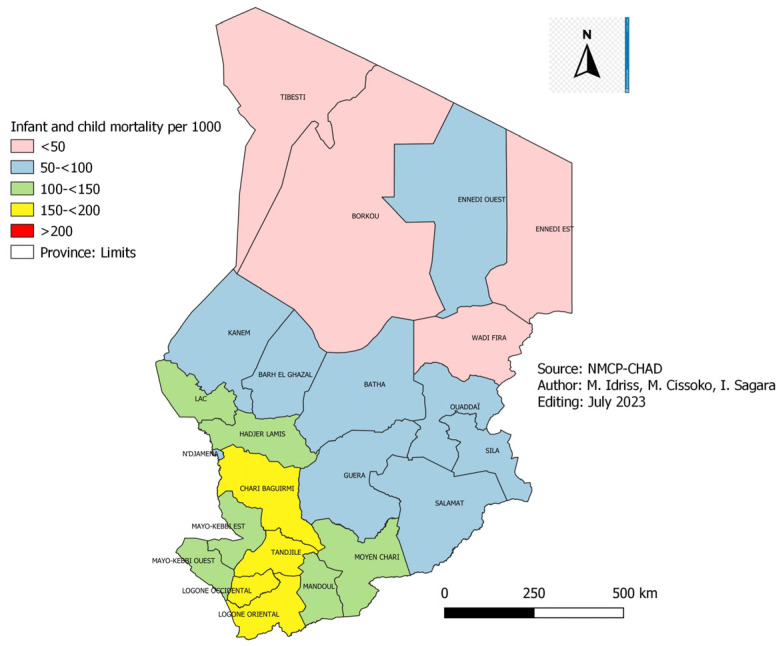
Infant and child mortality in 2019 in Chad.

**Figure 9 tropicalmed-08-00450-f009:**
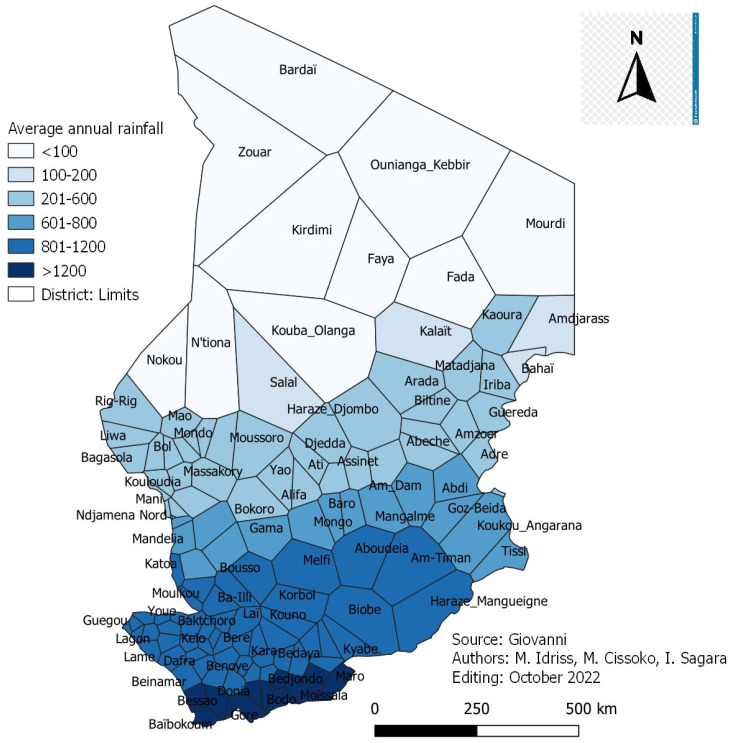
Annual average rainfall map over five (5) years (2017–2021) of health districts.

**Figure 10 tropicalmed-08-00450-f010:**
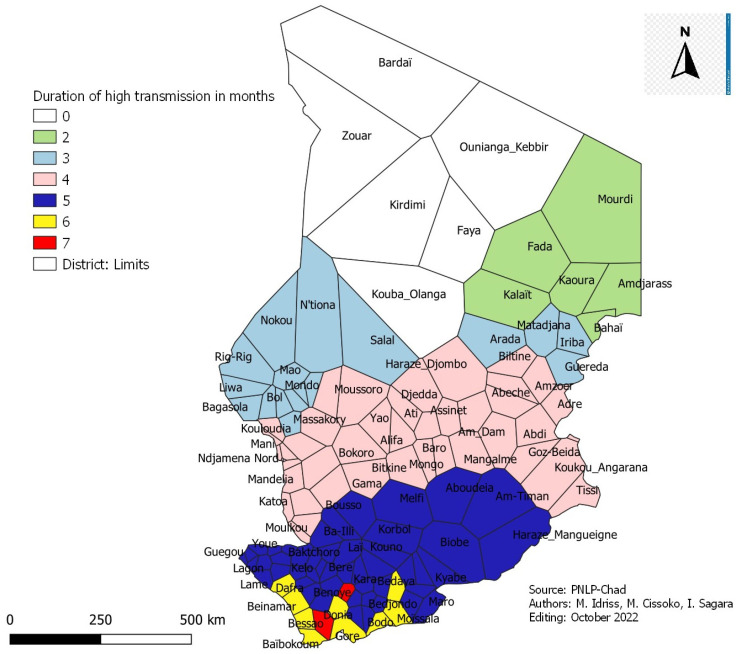
Map of districts with periods of high malaria transmission in months.

**Figure 11 tropicalmed-08-00450-f011:**
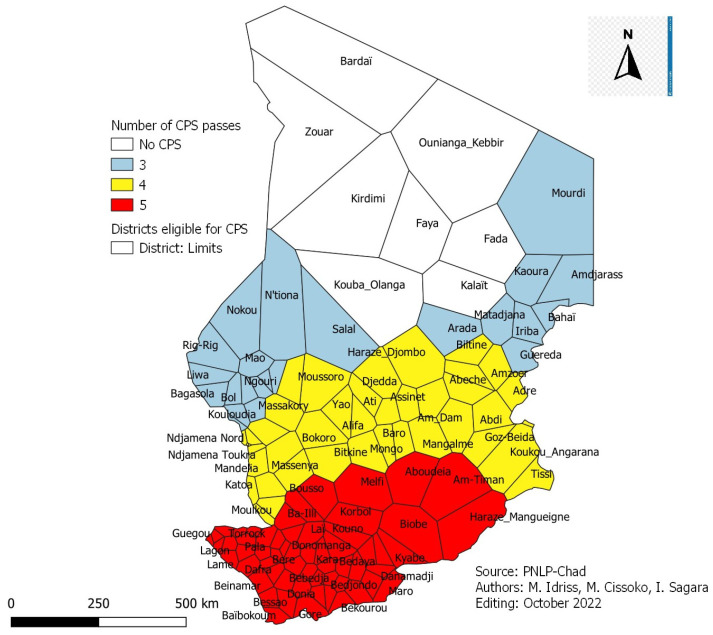
SMC-eligible districts and several passes.

**Figure 12 tropicalmed-08-00450-f012:**
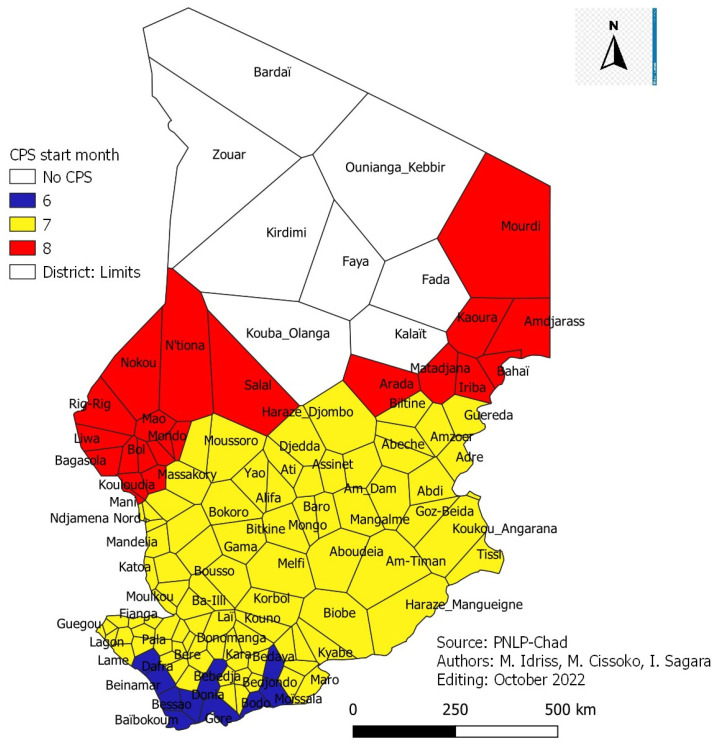
SMC starter map by the health district.

**Table 1 tropicalmed-08-00450-t001:** Summary of interventions by stratum.

Strata	Incidence Level	Interventions
Very low transmission	Incidence <10 cases per 1000	Case management, passive case detection, Active case detection, IPTp, Routine LLINs, IRS, Survence
	Incidence between 10 and <50 cases per 1000	Case management, IPTp, Routine LLINs, IRS: Reponse to epidemics
	Incedence between 50 and <100 cases per 1000	Case management, IPTp, Routine and campaign LLINs, IRS: Reponse to epidemics, SMC
Low transmission	Incidence between 100 and <250 cases per 1000	Case management, IPTp, Routine and campaign LLIN-PBO, SMC, IRS: Reponse to epidemics
Moderate transmission	Incidence between 250 and <450 cases per 1000	Case management, iCCM, IPTp, Routine and campaign LLIN-PBO, SMC
Strong transmission	Incidence greater than 450 cases per 1000	Case management, iCCM, IPTp, Routine and campaign LLIN-PBO, SMC, IRS

## Data Availability

The data is available in the database and in the survey reports used for this study with the authors.

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
