# Peer review of "Stratification and Adaptation of Malaria Control Interventions in Chad"

_tropicalmed, 2023, doi:10.3390/tropicalmed8090450_

Round 1

Reviewer 1 Report

The manuscript “Stratification and Adaptation of Malaria Control Interventions in Chad” is relevant and is in line with the scope of the Journal. However, the following modifications need to be addressed to make it publishable:

Comments 

Line 16. Entomological data (vector numbers)? Or data of positive human cases reported? 

Line 20. This is confusing, what does health districts mean in this sentence? Please re-write for better clarificarion. 

What does strata mean here? Neighborhoods, municipalities, states?

Line 26. This sentence sounds strange. Suggest English revision. 

Suggestion" Appropriate malaria control interventions were proposed based on the strata identified.  Stratification enables the country to target interventions in order to accelerate the reduction of the burden caused by malaria with a pre-elimination goal."

Line 32. When a Latinized genus name appears on its own, it should be written in italic.

Anopheles

Line 43. Please revise the way the numbers are witten. Comma, is the thousands separator used in most English-speaking countries, and a period is used to indicate the decimal place. 

It should be: 2,747.020

Line 44. It should be written as: 4,876.425.

Line 45. t should be written as: 2,228.512

Correct to: 1,548.039

Line 49. Please write the full name of the species. The word "and" does not need to be italicized.

Line 52. What does ss after the scientific name of the species mean? If this is a spelling mistake please correct it and delete the ss after the species name. 

If the idea was to refer to An. gambiae sensu stricto (s.s.). Than add ".".

Ex: An. gambiae s.s.

Line 56. If this is a acronym, please write the full name of the institution.

Line 59. Delete the word "for".

Line 98. The data were collected and analyzed for the 23 provinces or for the 129 health districts?

Line 107. The population data was per province or per district?

Line 129. The entire populaiton of Chad? Or the population per city (district)?

Line 175. Was the prevelance calculated per year?

Line 181. Were the distribution of parasites analyzed only for the year of 2017? Because all the other data were from 2017 to 2021 correct? 

Therefore all the data have to be from the same time period to allow comparasion. The analysis from 2018 to 2021 did not include parasite distribution?

Line 187. Change "‰" to "%".

Line 192. All data should be from the same time period (2017-2021) in order to allow comparasion. 

Entomological data was retrieved initiating in 2010, seven years before the year (2017) that all the other data was retrieved? What were these years compared to?

Line 231. This sentence needs correction. 

Suggestion: Considering the 129 health districts listed in 2021 in Chad, a total of......

Correct to: 1,548.039

Line 236. Delete this period before the sentence.

Line 240. Suggest placing the figures right underneath the text where they are mentioned.

Line 285. Figure 5 and 6.

Line 500. Correct Figure number.

Line 502. Correct figure number.

Author Response

comments

Line 16. Entomological data (numbers of vectors)? Or data on reported positive human cases?

Thank you very much for your comments which will improve the quality of this Manuscript. We have now specified that the entomological data come from studies carried out by the NMCP.

Line 20: This is confusing, what do the health districts mean in this sentence? Please rewrite for better clarification.

What do strata mean here? Neighbourhoods, municipalities, states?

We agree with your observation. It is health districts rather than strata and we have already changed this in the Manuscript.

Line 26. This sentence seems strange. Suggest a revision in English.

Suggestion "Appropriate malaria control interventions have been proposed according to the strata identified. Stratification allows the country to target interventions to accelerate the reduction of the burden caused by malaria with a goal of pre-elimination."

We agree with your suggestion and have incorporated it into the text.

Line 32. When a Latinised genus name appears alone, it should be written in italics.

Anopheles

We agree with your suggestion and have taken it into account throughout the document.

Line 43. Please revise the way the numbers are written. The comma is the thousands separator used in most English-speaking countries, and a full stop is used to indicate the decimal.

It should read: 2 747.020

Line 44. It should be written as follows: 4 876,425.

Line 45. t should read: 2,228.512

Correct to : 1 548,039

Thank you for your comment. We have followed the instruction of the editor to the authors related to the number and comma. We therefore levave number and comma as per the editor’s recommendation.

Line 49. Please write the full name of the species. The word "and" does not need to be italicised.

We agree with you. We have already inserted the full name of the species in the sentence.

Line 52. What does ss mean after the scientific name of the species? If this is a spelling error, please correct it and delete the ss after the species name.

If the idea was to refer to An. gambiae sensu stricto (ss). Then add ".

Ex: An. gambiae ss

This refers to An. gambiae sensu stricto (ss).

Line 56. If this is an acronym, please write the full name of the institution.

National Malaria Control Programme (NMCP)

Line 59. Delete the word "for".

We agree with you and have deleted the sentence.

Line 98. Were the data collected and analysed for the 23 provinces or for the 129 health districts?

For the 129 health districts.

Line 107. Were the demographic data by province or by district?

The demographic data for the EDST was by province.

Line 129. The entire population of Chad? Or the population by town (district)?

This is the population by health district.

Line 175. Was the prevalence calculated per year?

As this was a National Malaria Indicator Survey (NMIHS) in 2021, the prevalence was calculated during the survey period and therefore on the provincial or national population.

Line 181. Was the distribution of parasites analysed only for 2017? Because all other data were from 2017 to 2021 correct?

The data is from the NPIA carried out in 2017, so the analysis was only done for that year. There is a survey planned for 2022 but that is being carried out in 2023.

Consequently, all the data must come from the same period to allow comparison. The analysis from 2018 to 2021 did not include the distribution of parasites?

Line 187. Replace "‰" with "%".

OK, we've already replaced it.

Line 192. All data should be from the same period (2017-2021) to allow comparison.

We completely agree with you, except that the surveys are only carried out every five years.

The entomological data was collected in 2010, seven years before the year (2017) when all the other data was collected? What were these years compared to?

The National Malaria Control Programme conducts surveys every year to monitor the resistance of vectors to insecticides. We have used these data to show the monitoring of vector resistance to insecticides in Chad beyond 2017-2021, as this will provide information on vector resistance to insecticides since 2010.

Line 231. This sentence should be corrected.

Suggestion: Considering the 129 health districts identified in 2021 in Chad, a total of ......

We agree with your suggestion and have corrected it as suggested.

"A total of 1,548,039 cases of malaria were recorded in Chad's 129 health districts in 2021."

Correct to: 1 548,039

Line 236. Delete the full stop before the sentence.

Thank you for your comment. We have already deleted it.

Line 240. Suggest placing the figures just below the text where they are mentioned.

Line 285. Figures 5 and 6.

We agree with you and have corrected it.

Line 500. Figure number correct.

Thank you.

Line 502. Figure number correct.

Thank you very much!

Reviewer 2 Report

The information in this paper is very important and will be interesting to readers and programme managers in other countries who are trying to do the same. The effort to redo the stratification is appropriate and long overdue. I appreciate the use of both cross-sectional surveys and routinely reported incidence for stratification, and the relevant adjustment for diagnosis and attendance to improve the estimates, which is a relatively novel and welcome change.  I appreciate the attention to seasonality.

The use of geographic information is good although there are too many maps and their effect is diluted. Also the numbering of maps is incorrect. 

However some major changes are needed to improve the paper. In general, it needs to be cut and focused on stratification only.

1. I would like to know more about how the adjustment for diagnostics/attendance rate is done. The sentence lines 157-158 is not clear (if you are assuming 100% attendance how is that an adjustment?)  I would also like to know quantitatively what % difference each adjustment makes to the estimates. I suggest putting this information in tables by district in supplementary files, drop Figs 2 and 3, and only presenting maps for the crude and final adjusted incidence by district. 

2. It is confusing switching the geographical unit from province to district. I understand that the cross-sectional surveys only had province level information.  I suggest presenting all the province level information first and then go to district level.  The infant and child mortality results may be better in a table than two maps by province, one of which is all red (can you change the scales at least?)

3. I would like to see more data on the results of the 2017 survey. What was the sample of all age groups? How can the overall prevalence be 41% and the same for children under 5? (lines 41-42 and elsewhere. This is hard to believe. 

4. Section 3.10.1 does not belong in results and should be moved or removed.  I recommend to drop the parts about suggested interventions from the paper altogether as that is for discussion, and depends on cost and effectiveness of the suggested approaches (SMC, LLIN etc).  It would be a much stronger and clearer paper if it was only about the stratification leaving the rest for other paper(s).  The intervention choice by stratum (section 3.11 and related parts in Discussion section) need much more review and justification - I suggest remove.  This would leave room in Discussion for more review of the stratification methods and how this compares to what was done in past, and what is being done elsewhere. Including the cutoffs used to define high and low incidence. 

5. Similarly, I don't see the use of the entomology data or the climate info in stratification.  That could all be in another paper too which might relate these factors to the incidence. While important information areas, I don't see the relevance of them here and it comes across as just throwing anything and everything into the paper whether it is used in stratification or not. 

6. Ordering of the information needs tidying up. The attendance map (labeled Fig 153) should be given earlier as it relates to adjustments. Currently it comes between two sections on seasonality which belong together. 

7. The first sentence of introduction is over-dramatic.  Malaria is not the most widespread or most feared (that is subjective). What about TB, HIV, pneumonia, even COVID? Please tone it down. 

Its' comprehensible, but please get a native speaker to read and correct. 

Specie is not a word. 

I think facies should be facets.  

Author Response

  1. I would like to know more about how the adjustment for diagnoses/attendance rates is made. Sentence lines 157-158 are not clear (if you assume 100% attendance, how is that an adjustment?) I would also like to know quantitatively what % difference each adjustment makes to the estimates. I suggest putting this information in tables by district in supplementary files, deleting figures 2 and 3, and presenting only maps for gross and final adjusted incidence by district.

Thank you for your interest in this manuscript.

The adjustment to the biological confirmation rate and to the attendance rate was made as follows:

Crude incidence :

The crude incidence of malaria per 1000 person-years was calculated by dividing confirmed cases by the population times 1000.

Incidence adjusted for the rate of biological confirmation:

Given that not all suspected cases of malaria were tested for biological confirmation, we assumed that this calculated crude incidence did not reflect the true situation. To this end, we made an

adjustment using the biological confirmation rate.

The biological confirmation rate per health district was calculated by dividing confirmed cases by tested cases by 100.

The WHO recommends to countries that all suspected cases of malaria should be confirmed parasitologically before treatment is started. This means that all suspected cases must be 100% tested (suspected cases = tested cases). We therefore applied the biological confirmation rate to suspected cases to estimate the adjusted cases. These adjusted cases per 1000 of the population were used to determine the incidence adjusted for the biological confirmation rate.

Incidence adjusted for attendance rate :

The attendance rate was determined for each health district by dividing the total number of consultations for all diseases by the population times 100. This rate was used to calculate the adjusted cases per health district.

Cases adjusted to the attendance rate were determined by dividing confirmed cases by the attendance rate. These adjusted cases, divided by the population per 1000 people, were used to determine the incidence adjusted for the attendance rate.

If the attendance rate is 100%, there is no adjustment.

When the attendance rate is low, the adjustment becomes very important, and vice versa.

Incidence adjusted for biological confirmation rate and attendance rate:

This was calculated as follows:

Firstly, cases adjusted for the biological confirmation rate and the attendance rate were calculated by dividing cases adjusted for the biological confirmation rate by the attendance rate. These adjusted cases per 1000 population were used to determine the incidence adjusted for the biological confirmation rate and the attendance rate.

A table showing these data by health district is attached as an additional file.

We have also removed Figures 2 and 3 as requested.

  1. It is confusing to change the geographical unit from province to district. I understand that the cross-sectional surveys only had information at provincial level. I suggest presenting all the information at provincial level first, then moving to district level. The infant and child mortality results may be better in a table than two maps per province, one of which is all red (can we at least change the scales?).

These are the results of two Demographic and Health Surveys carried out in 2004 and 2019. The results were presented by province.

We agree with you about changing the scales to see the levels of mortality in the provinces. We have changed the scales on the maps below:

Figure 7: Infant and child mortality in 2004 in Chad.

Figure 8: Infant and child mortality in 2019 in Chad.

  1. I would like to see more data on the results of the 2017 survey. What was the sample of all age groups? How can the overall prevalence be 41% and the same for children under 5 (lines 41-42 and elsewhere. This is hard to believe.

The results presented here are from the 2017 NPTS, details of which can be found on the validated 2017 NPTS report which will be attached as an additional item (p.83-89).

  1. Section 3.10.1 does not belong in the results and should be moved or deleted. I recommend deleting the suggested interventions sections of the paper altogether as this is for discussion and depends on the cost and effectiveness of the suggested approaches (CPS, MILD, etc.). It would be a much stronger and clearer paper if it dealt only with stratification, leaving the rest for other papers. The choice of intervention by stratum (section 3.11 and related parts in the Discussion section) requires much more consideration and justification - I suggest deleting it. This would leave room in the discussion for further consideration of stratification methods and how this compares to what has been done in the past and elsewhere. Including the thresholds used to define high and low incidence.

The thresholds we have used to define high and low malaria incidence are those of the WHO Malaria Elimination Framework, 2017.

Apart from the thresholds indicated by the WHO as mentioned above, there are no standard thresholds for defining high and low incidences for all countries, depending on the actual incidences of malaria, each country defines its threshold.

For the interventions, we have modified the text to indicate that these are suggested interventions. They are relevant because the WHO's recommend for malaria-endemic countries to perform stratification based on their gudlenine so that malaria interventions are adapted to the zones in a given country, for greater efficiency in the fight against malaria. The interventions suggested here are only indicative and can therefore vary if necessary in the field, because stratification is dynamic (changing epidemiology of malaria in time and space) while resources may also be limited to implement sometimes according to the stratification plan even if this stratification is a tool to make the choice according to the scale. This clarification is now made in the manuscript and we thank you for your comments on this aspect.

Also, one of your fellow reviewer’s #4 appreciated the proposed interventions following this stratification and asked that a table be made, which was done.

  1. Similarly, I don't see the use of entomological data or climatic information in stratification. All of this could also be included in another article that could link these factors to incidence. Although important areas of information, I don't see their relevance here and it gives the impression of throwing anything and everything into the paper, whether it is used in stratification or not.

We agree with you to remove some sections in the document especially sections 3.9.1, 3.9.2 and 3.12.2 including maps 10, 12, 17. However, we have kept climatic information such as rainfall map to help understand the stratification.

  1. The information must be arranged in order. The frequentation map (labelled Fig 153) should be given earlier as it concerns adjustments. At present, it is between two sections on seasonality that go together.

We agree with you that the order of the information should be reviewed. Section 3.10.2 is moved back after the impact adjustments and becomes section 3.7.

The Fig 13 map becomes Fig 3.

  1. The first sentence of the introduction is too dramatic. Malaria is not the most widespread or the most feared (this is subjective). What about tuberculosis, HIV, pneumonia and even COVID? Please tone it down.

We agree with your observation. We have reviewed the wording of the first introductory sentence: "Malaria is a major public health problem worldwide, particularly in sub-Saharan Africa."

Reviewer 3 Report

The information presented in this article is a highly valuable contribution to our understanding of the malaria transmission situation in Chad. It is well written. The data collection methods and analyses conform to standard practices and recommendations of the WHO. I would recommend its publication with only a few minor modifications.

The manuscript needs another proofread to correct a few typos, for example: Line 341 - "fenitrition" should be corrected to "fenitrithion."  Some of the figures capitalize species names for malaria parasites. It is standard practive to capitalize only the genus, while species and subspecies are lower case (i.e. Plasmodium falciparum). 

Figure 8 might be more informative if the color key was altered to reveal differences in child and infant mortality rates in 2004.  Certainly, it is striking to see the map in 2004 compared to the present day, but readers might also be interested in visualizing the regional differences that I'm sure were present in 2004.

The entomological data is a bit lacking in this manuscript with only binary (presence/absence) or qualitative information presented. If quantitative data for insecticide resistance from bioassay results is available it should be included. Otherwise, I might suggest that the claim of presence or absence of resistance cannot be objectively assessed by the reader. If quantitative data are not available, I would recommend removing this section. I don't think that removal of some of this qualitative entomological info would hurt the quality of this manuscript. Perhaps the descriptive data on the whereabouts of vector species could be retained.

My final observation is that this manuscript as a bloated feel to it. The manuscript is much longer than it need to be to effectively present its salient findings, namely the stratification of malaria risk. I think that the message it intends to convey could be made much more succinctly by removing up to 20% of the figures. Perhaps the authors could review their figures and decide which ones could be removed. I think that would create a more focused and accessible document for readers. This is just a suggestion.

All in all, I feel that this is valuable and useful work that deserves rapid publication. It is well organized and well written, and just maybe a bit more comprehensive than necessary.

Very good writing!  No recommendation required other than a reread to check for a few typographical errors (misspellings, punctuation, capitalization).

Author Response

The manuscript needs another proofread to correct a few typos, for example: Line 341 - "fenitrition" should be corrected to "fenitrithion."  Some of the figures capitalize species names for malaria parasites. It is standard practive to capitalize only the genus, while species and subspecies are lower case (i.e. Plasmodium falciparum). 

Thank you for your comments. We have taken all your comments into account throughout the document.

Figure 8 might be more informative if the color key was altered to reveal differences in child and infant mortality rates in 2004.  Certainly, it is striking to see the map in 2004 compared to the present day, but readers might also be interested in visualizing the regional differences that I'm sure were present in 2004.

Like reviewer 2, we agree with you that the scales should be changed to show the levels of mortality in the provinces. We have changed the scales in the manuscript (Figures 7 and 8).

The entomological data is a bit lacking in this manuscript with only binary (presence/absence) or qualitative information presented. If quantitative data for insecticide resistance from bioassay results is available it should be included. Otherwise, I might suggest that the claim of presence or absence of resistance cannot be objectively assessed by the reader. If quantitative data are not available, I would recommend removing this section. I don't think that removal of some of this qualitative entomological info would hurt the quality of this manuscript. Perhaps the descriptive data on the whereabouts of vector species could be retained.

Quantitative data on the resistance of vectors to insecticides derived from the results of biological trials are not available, except for descriptive data on the location of vector species. We therefore agree with you that this section should be deleted.

My final observation is that this manuscript seems bloated. The manuscript is much longer than necessary to effectively present its main results, namely the stratification of malaria risk. I think its intended message could be made much more succinct by deleting up to 20% of the figures. Perhaps the authors could review their figures and decide which ones could be deleted. I think this would create a more focused and accessible document for readers. This is just a suggestion.

We agree with you and have removed some sections (entomology section).

Overall, I think it's a valuable and useful piece of work that deserves to be published soon. It is well organised and well written, and perhaps a little more comprehensive than necessary.

Comments on the quality of the English language

Very good writing! No recommendations required other than proofreading to check a few typographical errors (spelling mistakes, punctuation, capitalisation).

Reviewer 4 Report

This paper aimed to estimate the incidence of Malaria one of the main cause of death in Chad, stratify by patient characteristics and environment data to be inform Malaria control measures policies. The paper show the strong link between malaria incidence and climate conditions. The incidence varied according to stratification and adjustment to laboratory confirmation rate or attendance rate.

Comments :

1)      The authors described in the methods of calculation of the crude incidence but the methods of calculation of the adjusted incidences are not clearly described. The authors needed to specify how they estimated the incidence adjusted by laboratory confirmation rate or attendance rate.  Did they used a poisson regression model or other regression techniques? The results presented for the adjusted incidence looks more like corrected incidences than adjusted incidences. The description of the methods and calculation may clarify this point.

2)      The incidence adjusted for biological confirmation rate is based on the estimation of suspected cases of malaria. How are defined the suspected cases of malaria? Are they based on clinical symptoms, if yes which symptoms and how are they collected?

3)      For the data quality checking the reference period was 2021 but the period of the visits of the 66 health facilities were not specified.

4)      One the interesting results of this paper is the adaptation of the malaria control measure using the incidence estimated. The authors need to better describe the targeting interventions to be evaluated and their combinations. A table for example may be interesting. For the results 3.10.1 perhaps a decision algorithm should be explain the authors different choices about interventions combination and incidence stratification zones.  

Author Response

Comments :

1)  The authors describe in the methods for calculating crude incidence but the methods for calculating adjusted incidence are not clearly described. The authors were asked to specify how they estimated the incidence adjusted by the laboratory confirmation rate or the attendance rate. Did they use a Poisson regression model or other regression techniques? The results presented for the adjusted incidence look more like corrected incidences than adjusted incidences. The description of the methods and calculations may shed some light on this point.

Thank you for your interest in this manuscript.

It is indeed a corrected incidence.

We detailed the correction method in the manuscript.

The calculation methods we used to determine the impact adjustments are as follows:

Crude incidence:

The crude incidence of malaria per 1000 person-years was calculated by dividing confirmed cases by the population times 1000.

Incidence adjusted for the rate of biological confirmation of malaria:

Given that not all suspected cases of malaria were tested for biological confirmation, we assumed that this crude incidence calculated on the basis of confirmed cases of malaria did not reflect the true situation. To this end, we made an adjustment using the biological confirmation rate.

The rate of biological confirmation per health district was calculated by dividing confirmed cases by tested cases by 100.

The WHO recommends to countries that all suspected cases of malaria should be diagnosed parasitologically before treatment is started. This means that all suspected cases must be tested at 100% (suspected cases = tested cases). We therefore applied the biological confirmation rate to suspected cases to estimate the adjusted cases. These adjusted cases per 1000 of the population were used to determine the incidence adjusted for the biological confirmation rate.

Incidence adjusted for attendance rate :

The attendance rate was determined for each health district by dividing the total number of consultations for all diseases over the population by 100. This rate was used to calculate the adjusted cases per health district.

Cases adjusted to the attendance rate were determined by dividing confirmed cases by the attendance rate. These adjusted cases, divided by the population per 1000 person-years, were used to determine the incidence adjusted for the attendance rate.

Incidence adjusted for the rate of biological confirmation and the attendance rate:

This was calculated as follows:

Firstly, cases adjusted to the biological confirmation rate and the attendance rate were calculated by dividing cases adjusted to the biological confirmation rate by the attendance rate. These adjusted cases per 1000 population were used to determine the incidence adjusted for the biological confirmation rate and the attendance rate.

A table presenting these data by health district is attached as an additional file.

2) The incidence adjusted for the rate of biological confirmation is based on the estimate of suspected cases of malaria. How are suspected cases of malaria defined? Are they based on clinical symptoms, and if so, which symptoms and how are they collected?

The incidence adjusted for the biological confirmation rate is estimated by multiplying suspected malaria cases by the biological confirmation rate.

Suspected malaria case:

Simple malaria: The patient presents with the following symptoms: fever, headache, body aches, chills, digestive problems in children. A parasitological examination is requested to confirm the diagnosis.

Severe malaria: Hyperthermia or hypothermia, neurological disorder, etc. A parasitological examination confirms the diagnosis.

For data collection, a curative care register and a monthly malaria report form are made available to health facilities for monthly data collection.

Data is collected at three levels

Peripheral level (health centre and district hospital)

Intermediate level (provincial hospitals and medical practices/clinics)

Central level (referral hospitals and CHU)

All health facility data are compiled monthly in a National Malaria Database managed by the NMCP.

3) For the verification of data quality, the reference period was 2021 but the period of the visits to the 66 health facilities was not specified.

The quality of the data from the 66 health facilities was checked in the second half of 2020 and 2021 (periods of high malaria transmission in Chad). Although morbidity data were collected from 2017 to 2021, only two years were selected for data verification.

4) One of the interesting results of this article is the adaptation of the malaria control measure using estimated incidence. The authors should better describe the targeting interventions to be evaluated and their combinations. A table, for example, might be useful. For results 3.10.1, a decision algorithm should perhaps explain the different choices made by the authors concerning the combination of interventions and the incidence stratification zones.

We agree with your proposal. We have inserted a table N°1 summarising the interventions by stratum. The choice was made according to the level of endemicity of each stratum.